# miR-196a Promotes Proliferation of Mammary Epithelial Cells by Targeting *CDKN1B*

**DOI:** 10.3390/ani13233682

**Published:** 2023-11-28

**Authors:** Guanhe Chen, Wenqiang Sun, Yuchao Li, Mengze Li, Xianbo Jia, Jie Wang, Songjia Lai

**Affiliations:** College of Animal Science and Technology, Sichuan Agricultural University, Chengdu 611130, China; cgh_0132@163.com (G.C.); wqsun2021@163.com (W.S.); lichao_1116@163.com (Y.L.); 15706065790@163.com (M.L.); jaxb369@sicau.edu.cn (X.J.); wjie68@163.com (J.W.)

**Keywords:** dairy cow, heat stress, miR-196a/*CDKN1B* axis, proliferation arrest

## Abstract

**Simple Summary:**

The physiological status of a dairy cow’s udder is significantly impacted by heat stress (HS). Therefore, improving the physiological function of dairy cow udders under HS conditions is important to improve animal welfare and dairy production efficiency. In the present study, we found that HS inhibited the proliferation of bovine mammary epithelial cells (BMECs) and significantly downregulated the expression of endogenous miR-196a. Furthermore, overexpression of miR-196a could alleviate the inhibitory effect of HS on BMEC proliferation. Additional data confirmed that miR-196a promoted BMEC proliferation. Subsequently, we identified *CDKN1B* as a target gene of miR-196a. The effect of miR-196a on BMEC proliferation was reversed by *CDKN1B*. In summary, we found that miR-196a could promote BMEC proliferation by targeting *CDKN1B*. These findings can provide a reliable reference for alleviating the inhibition of BMEC proliferation caused by HS.

**Abstract:**

Heat stress (HS) has become one of the key challenges faced by the dairy industry due to global warming. Studies have reported that miR-196a may exert a role in the organism’s response to HS, enhancing cell proliferation and mitigating cellular stress. However, its specific role in bovine mammary epithelial cells (BMECs) remains to be elucidated. In this study, we aimed to investigate whether miR-196a could protect BMECs against proliferation arrest induced by HS and explore its potential underlying mechanism. In this research, we developed an HS model for BMECs and observed a significant suppression of cell proliferation as well as a significant decrease in miR-196a expression when BMECs were exposed to HS. Importantly, when miR-196a was overexpressed, it alleviated the inhibitory effect of HS on cell proliferation. We conducted RNA-seq and identified 105 differentially expressed genes (DEGs). Some of these DEGs were associated with pathways related to thermogenesis and proliferation. Through RT-qPCR, Western blotting, and dual-luciferase reporter assays, we identified *CDKN1B* as a target gene of miR-196a. In summary, our findings highlight that miR-196a may promote BMEC proliferation by inhibiting *CDKN1B* and suggest that the miR-196a/CDKN1B axis may be a potential pathway by which miR-196a alleviates heat-stress-induced proliferation arrest in BMECs.

## 1. Introduction

With the ongoing shifts in the global climate, HS has emerged as a significant challenge for the dairy cattle production industry [1]. The physiological state of the udder significantly influences milk productivity under HS [2]. Changes in the number or functional capacity of mammary epithelial cells correlate with reductions in milk production. HS is observed to inhibit cell growth and alter structural proteins, membrane permeability, and metabolism within these cells [3]. Concurrently, some studies have reported adverse effects of HS on the viability and cell cycle phase of BMECs [4].

In a previous HS experiment with rabbits, our team identified 23 differentially expressed miRNAs, including miR-196a, between tissues under HS and a control condition [5]. This information suggests that miR-196a, identified in the study, could play a role in modulating cellular physiological changes during HS. Previous research has shown that miR-196a can enhance the proliferation of cancer cells by targeting *FOXO1* [6]. miR-196a’s ability to downregulate *HOXA5* was found to stimulate nonsmall lung cancer and gastric cancer cell proliferation [7,8]. In breast cancer, *UBE2C* was identified as a new target gene of miR-196a, and its upregulation by miR-196a promoted cell proliferation [9]. In swine, miR-196a was found to promote cell proliferation in immature porcine Sertoli cells by binding the 3′UTR of *RCC2* and *ABCB9* [10]. Meanwhile, miR-196a was also found to be potentially involved in regulating fat deposition and muscle differentiation and development in cattle [11,12]. In addition, miR-196a has been recognized for enhancing cellular stress resistance [13]. Nevertheless, the regulatory function of miR-196a in BMECs remains elusive. Thus, exploring the potential roles of miR-196a in BMECs is a valuable endeavor.

Modifications in cell cycle progression play a pivotal role in regulating cellular proliferation. In eukaryotic cells, the progression of the cell cycle is overseen by the sequential expression of cell cycle proteins. These proteins activate the transcription of specific genes corresponding to different cycle-dependent kinases (CDKs), thereby promoting cell cycle progression [14]. Additionally, cell cycle progression is influenced by CDK inhibitors. Overexpressing these genes might hinder cell proliferation [15]. Among these inhibitors, *CDKN1B* stands out. An upregulation of *CDKN1B* has been observed to inhibit cell proliferation. Numerous studies have shown that *CDKN1B* is integral to the cellular response to HS [16]. *CDKN1B*, being an oxidative-stress-related gene, becomes upregulated during HS. This upregulation is vital for the selective elimination of protein aggregates caused by HS, thus effectively reducing cellular proteotoxic stress [17]. 

Our objective for this study was to investigate whether miR-196a could protect BMECs against proliferation arrest induced by HS and explore its potential underlying mechanism. In the present study, we measured a significant downregulation in miR-196a expression under HS compared to BMECs cultured at 37 °C (control). Further functional analyses suggested that miR-196a can act as a promoter of BMEC proliferation. A luciferase reporter assay corroborated that miR-196a directly targets *CDKN1B* in BMECs. Moreover, data from the rescue experiment demonstrated that miR-196a promotes BMEC proliferation by binding to *CDKN1B*.

## 2. Materials and Methods

### 2.1. Cell Culture

BMECs were seeded and cultured in DMEM/F12 medium (Meilunbio, Dalian, China, MA0214) supplemented with 10% fetal bovine serum (FBS, Gibco, San Jose, CA, USA, 10099141C) and 1% penicillin–streptomycin (Solarbio, Beijing, China, P1400). They were kept in an incubator (Thermo Fisher Scientific, San Jose, CA, USA) at 37 °C and 5% CO_2_. For further experiments, the cells were subcultured or seeded in 6-, 12-, 24-, and 96-well plates (NEST, Wuxi, Jiangsu, China, 703002, 712002, 702002, 701002) until they reached about 80% cell density. 

### 2.2. Heat Stress Exposure

The temperature of 42 °C applied to the HS model was based on a previous study conducted in our laboratory [18]. Firstly, BMECs were seeded at a density of 1.25 × 10^6^/well in 6-well plates or 5 × 10^4^/well in 96-well plates and cultured in a Thermo Fisher Scientific incubator (Thermo Fisher Scientific, San Jose, CA, USA) at 37 °C and 5% CO_2_ environment until reaching a cell density of 50–60%. At this point, cells were divided into two groups: the control group continued to be cultured in the same incubator, while the HS group was transferred to another incubator at 42 °C and maintained under 5% CO_2_. We collected both groups of cells at time points of 0, 6, 12, and 24 h.

### 2.3. Transfection

We used a Lipofectamine™ 3000 transfection kit (Invitrogen, Carlsbad, CA, USA) to transfect miR-196a mimic (mimic), miR-196a inhibitor (inhibitor), miR-NC, inhibitor-NC (I-NC), PCDNA, and PCDNA-CDKN1B at a cell density reaching 50–60%. After 6 h, the culture medium was replaced with fresh medium. The RNA oligos were synthesized by Sangon Biotech Co., Ltd. (Shanghai, China). The miRNA mimic is a chemically synthesized mature double-stranded miRNA that enhances endogenous miRNA function. It consists of a sequence corresponding to the target miRNA mature body sequence and a complementary sequence to the miRNA mature body sequence. The miRNA inhibitor is a chemically synthesized mature single-stranded miRNA with methoxy modification designed to specifically target and efficiently inhibit endogenous miRNA activity in cells, enabling functional deficiency research on specific miRNAs. miR-NC and I-NC served as negative controls, respectively. Appendix A shows detailed RNA oligo sequences.

### 2.4. RNA Extraction and Real-Time Quantitative PCR (RT-qPCR)

Total RNA and miRNA were extracted using TRIpure reagent (Aidlab Biotechnologies, Beijing, China #RN0102) and the RNAmisi micropurification kit (Aidlab Biotechnologies, Beijing, China, #RN0501), respectively, following the manufacturer’s instructions. The concentration and quantity of total RNA were measured using NanoDrop 2000 (Thermo Fisher Scientific, San Jose, CA, USA).

Subsequently, mRNA and miRNA were reverse transcribed into cDNA using the HiScript III RT Super Mix for qPCR (+gDNA wiper) reagent kit (Vazyme, Nanjing, Jiangsu, China, #R323–01) and the stem-loop reverse transcription primer and miRNA 1st strand cDNA synthesis kit (by stem-loop) (Vazyme, Nanjing, Jiangsu, China, #MR101–02), respectively, following the instructions provided in the manual. ChamQ SYBR qPCR Master Mix (Vazyme, Nanjing, Jiangsu, China, #Q311–02) and miRNA Universal SYBR Master Mix (Vazyme, Nanjing, Jiangsu, China, #MQ101–02) were used to quantify mRNA and miR-196a expression, respectively. The Bio-Rad CFX96 real-time PCR instrument (Bio-Rad, Hercules, CA, USA) was used for RT-qPCR reactions.

The RT-qPCR conditions included an initial step at 95 °C for 5 min, followed by 40 cycles of denaturation at 95 °C for 15 s, annealing at 60 °C for 15 s, and extension at 72 °C for 20 s. We performed a thermal denaturation cycle to determine dissociation curves for verifying the specificity of PCR amplification. Each sample was tested in triplicate. All quantitative primers were designed using Primer 5.0 and synthesized by Tsingke Biotechnology (Beijing, China). Appendix A provides details on primers for RT-qPCR. The CQ values of *β-actin* and U6 small nuclear RNA were used as internal references to calculate gene mRNA expression levels and the relative abundance of miR-196a, respectively. The relative expression levels of genes and miR-196a were calculated using the 2^−ΔΔCt^ method. 

### 2.5. Bioinformatics Analysis

The initial raw data in Fastq format were processed using the Fastp software [19]. The reference genome and gene model annotation files were obtained from a genome website https://www.ncbi.nlm.nih.gov/datasets/taxonomy/9913/ (accessed on 11 August 2022) and indexed using Hisat2 v2.0.5 http://daehwankimlab.github.io/hisat2/ (accessed on 13 August 2022) [20]. The paired-end clean reads were then aligned to the indexed genome using the same software. We used featureCounts v1.5.0-p3 https://subread.sourceforge.net/featureCounts.html (accessed on 14 August 2022) [21] to count the reads mapped to each gene. Differential expression analysis was conducted using the DESeq2 R package version 1.20.0 https://bioconductor.org/packages/release/bioc/html/DESeq2.html (accessed on 15 August 2022) [22]. To perform Gene Ontology (GO) and Kyoto Encyclopedia of Genes and Genomes (KEGG) enrichment analysis of the differentially expressed genes, we used the clusterProfiler R package version 4.10.0 https://bioconductor.org/packages/release/bioc/html/clusterProfiler.html (accessed on 15 August 2022) [23].

### 2.6. Western Blotting

Total cellular protein was collected from BMECs using the ProteinExt^®^ mammalian total protein extraction kit (TransGene Biotech, Beijing, China, #DE101–01). The protein concentration of each sample was determined using a Bradford protein assay kit (Novoprotein, Shanghai, China, #PA001–01A) according to the manufacturer’s instructions. For electrophoresis, 30 µg of protein from each sample was loaded onto a 12% SDS-PAGE gel. The corresponding primary antibodies were incubated with the membranes at 4 °C for 8 h, followed by a 2 h incubation with secondary antibodies (Goat Anti-Rabbit IgG H&L (HRP), Zen Bioscience, Chengdu, China). The membranes were washed three times and developed using chemiluminescence in a Touch Imager Pro (e-BLOT Life Science Co., Ltd. Shanghai, China). The primary antibodies PCNA, CDKN1B, and β-actin were purchased from Proteintech (Wuhan, Hubei, China). β-actin served as the internal reference for normalization.

### 2.7. Luciferase Reporter Assay

The TargetScan Human 8.0 https://www.targetscan.org/vert_80/ (accessed on 6 September 2022) [24] and miRDB https://mirdb.org (accessed on 13 September 2022) [25] were used to predict miR-196a-related target genes. The potential binding site between miR-196a and *CDKN1B* was predicted using TargetScan Human 8.0 https://www.targetscan.org/vert_80/ (accessed on 15 September 2022) [24]. Fragments of 3′UTR containing potential binding sites of *CDKN1B*, namely, the wild type (WT) and mutant type (MUT), were cloned into the primGLO dual-luciferase reporter vector. HEK-293T cells were seeded into 24-well plates and then co-transfected with either the mimic or miR-NC, along with either the pmirGLO-WT or pmirGLO-MUT vectors. All transfections were conducted using the Lipofectamine™ 3000 transfection kit (Invitrogen, Carlsbad, CA, USA) following the manufacturer’s instructions. After 36 h of co-transfection, luciferase activities were measured using the dual-luciferase reporter assay system. The Renilla luciferase activities were used to normalize relative reporter activities.

### 2.8. EDU Assay

BMECs were seeded in 96-well plates and transfected when the cell density reached approximately 60%. The Beyoclick EDU-555 cell proliferation assay kit (Beyotime Biotechnology, Shanghai, China, #C0075S) was used for the EDU assay following the manufacturer’s instructions. After 36 h of transfection, an equal volume of EDU working solution was added to the wells and incubated for 1 h. DAPI (Solarbio, Beijing, China, C0065) was used to stain the cell nuclei and help observe the ratio of cell proliferation. The stained cells were observed under a fluorescence microscope (Olympus, Tokyo, Japan) with a magnification of 100 times, allowing the cell nuclei to appear blue and the proliferating cells to appear red. Photographs were captured using an inverted fluorescence microscope (Olympus, Tokyo, Japan) under identical visual field conditions. Subsequently, the images were analyzed using ImageJ 1.51 software (National Institutes of Health, Bethesda, MD, USA).

### 2.9. CCK-8 Assay

BMECs were seeded in 96-well plates and then transfected. After transfection, we added 10 µL of CCK-8 solution (Beyotime Biotechnology, Shanghai, China, #C0037) to each well and incubated them for 2 h. The optical density (OD) value at 450 nm was measured using a Varioskan LUX spectrophotometer (Thermo Scientific, Waltham, MA, USA). The obtained absorbance values were subsequently plotted and analyzed using GraphPad Prism 8 software (GraphPad, San Diego, CA, USA).

### 2.10. Statistical Analysis

All data were analyzed using Prism 8 software (GraphPad, San Diego, CA, USA). Statistically significant difference between the two groups was determined using Student’s *t*-test. For multiple comparisons, the one-way analysis of variance (ANOVA) method was employed. The results are presented as the mean ± SEM, and each independent experiment was performed in triplicate. Statistical significance was indicated by *p* < 0.05.

## 3. Results

### 3.1. Heat Stress Downregulates miR-196a Expression and Suppresses BMEC Proliferation

A prior study highlighted elevated expression levels of miR-196a in the mammary gland compared to other tissues [26]. To delve into the tissue-specific expression of miR-196a, we performed RT-qPCR to assess the expression patterns of miR-196a across various cow tissues, including the heart, liver, spleen, lung, kidney, adipose, mammary gland (mammary epithelial cells versus stroma), and skin. Figure 1A illustrates that miR-196a expression differed across these tissues. Specifically, miR-196a exhibited high expression in the kidney and mammary gland, moderate expression in the skin and adipose tissue, and minimal expression in the heart and ovarian tissues. These findings suggest that miR-196a may play an important regulatory function in the mammary glands of dairy cows.

One of our previous studies suggested that miR-196a may be involved in regulating the alteration of the physiology of the animal organism under HS [5]. We established an HS model using BMECs to investigate whether miR-196a regulates the mammary gland’s physiological response to HS in dairy cows. RT-qPCR was employed to determine the expression patterns of *HSPB1* and *HSP90AA1* under HS conditions at 42 °C. Notably, Figure 1B,C shows that the expression levels of both *HSPB1* and *HSP90AA1* were substantially elevated in BMECs exposed to 42 °C compared to the initial culture stage (0 h). Additionally, *HSPB1* and *HSP90AA1* expression levels rose with prolonged exposure to HS at intervals of 0, 6, and 12 h. Subsequently, we assessed fluctuations in miR-196a expression under HS at 0, 6, 12, and 24 h using RT-qPCR. We identified a significant decline in miR-196a expression post HS exposure, with its expression being the least at 24 h. The miR-196a expression in the control group showed no significant change (Figure 1D). Furthermore, we noticed a pronounced reduction in BMECs’ proliferative activity during HS (Figure 1E). Nevertheless, increasing miR-196a expression appeared to alleviate the suppressive effect of HS on cellular proliferation (Figure 1F). 

### 3.2. mRNA-Seq and Bioinformatics Analysis Were Used to Explore the Potential Functions of miR-196a

To investigate the biological function of miR-196a in BMECs, we conducted transcriptome profiling on BMECs transfected with miR-196a mimic and miR-NC. We constructed six cDNA libraries with three replicates for each group (mimic and miR-NC) and sequenced them using the Illumina NovaSeq platform (Illumina, San Diego, CA, USA). The sequencing produced 150 bp paired-end reads, yielding over 4.2 million raw reads for each sample. After discarding low-quality and unclear reads, a minimum of 97.55% of raw reads remained as clean reads in every sample. These clean reads were aligned to the *Bos taurus* reference genome, with mapping rates ranging from 96.02 to 97.08% in individual samples (Appendix A). These outcomes indicate the sequencing data’s reliability and suitability for subsequent bioinformatics processing.

To pinpoint genes with altered expression due to miR-196a overexpression, we identified 105 DEGs (Appendix A). These included 48 upregulated genes, such as *MMP3*, *EDN1*, *ATP5ME*, *NDUFAF8*, and *FGF21*, and 57 downregulated genes, such as *FOXO1*, *GADD45A*, and *SOCS2*, using padj < 0.05 and |log2foldchange|≥ 1 as criteria (Figure 2A). The genes *ATP5ME*, *NDUFAF8*, and *FGF21* are reportedly associated with cellular energy metabolism and thermogenesis, while *FOXO1*, *GADD45A*, and *SOCS2* are vital in regulating cell proliferation. Five DEGs were randomly selected based on the following criteria: |log2foldchange| > 1, padj < 0.05, FPKM > 50, and read count > 20. Their expression was verified with RT-qPCR to confirm the RNA-seq results’ reliability, which showed a similar trend to that of sequencing (Figure 2B). For a deeper understanding of DEG roles, we conducted GO and KEGG enrichment analyses. We categorized the 105 DEGs into three primary groups: biological process (BP), cellular component (CC), and molecular function (MF) (Appendix A). Figure 2C shows the top 10 GO terms from each category impacted by miR-196a overexpression. Most DEGs were grouped under the regulation of multicellular organism processes (BP), extracellular regions (CC), and signaling receptor binding (MF). Additionally, the DEGs were annotated across 171 pathways in the KEGG analysis (Appendix A). Figure 2D presents the top 20 KEGG pathways influenced by miR-196a overexpression. Most DEGs were enriched in cellular senescence and transcription misregulation in cancer, followed by nine equal-sized pathways, such as TNF signaling and JAK-STAT signaling. Notably, numerous studies have linked many top 20 pathways, including calcium signaling, cAMP signaling, P53 signaling, JAK-STAT signaling, and others, to cell proliferation regulation [27,28,29,30,31]. These data suggest that miR-196a’s primary role in BMECs may be to regulate cell proliferation.

### 3.3. Effect of miR-196a on Cell Proliferation of BMECs

To assess miR-196a’s impact on BMEC proliferation, we employed the CCK-8 assay to gauge cellular viability. We observed a marked enhancement in cell viability in BMECs treated with the miR-196a mimic relative to miR-NC (Figure 3A). On the contrary, BMECs transfected with the miR-196a inhibitor relative to the I-NC group were significantly decreased in cell viability (Figure 3B). Furthermore, upon conducting the EDU assay, we determined that the rate of EDU-positive cells in the mimic group was significantly higher than in the miR-NC group (Figure 3C,D). Meanwhile, the proportion of EDU-positive cells was considerably diminished in BMECs treated with the miR-196a inhibitor compared to the I-NC group (Figure 3E). Additionally, we analyzed the relative expression levels of proliferation-associated genes, namely, *PCNA*, *CDK2*, and *MCM3*, in BMEC post-transfection with miR-196a mimic, miR-196a inhibitor, miR-NC, and I-NC. *PCNA*, *CDK2*, and *MCM3* transcript abundance increased significantly when miR-196a was overexpressed. Conversely, inhibiting miR-196a markedly reduced their transcript levels (Figure 3F–H). miR-196a’s role in promoting BMEC proliferation was further substantiated by assessing the protein expression of *PCNA* (Figure 3I). *PCNA* protein levels were either substantially increased or decreased in BMECs transfected with a miR-196a mimic or inhibitor compared to cells with negative controls, respectively. Collectively, these findings suggest that miR-196a enhances BMEC proliferation.

### 3.4. CDKN1B Is a Direct Target Gene of miR-196a in BMECs

To investigate the underlying mechanisms through which miR-196a modulates the proliferation of BMECs, we used TargetScan Human 8.0 and miRDB to computationally predict potential target genes regulated by miR-196a (Appendix A). By integrating these predictions with downregulated genes identified in our RNA-seq analysis, *CDKN1B* emerged as a promising candidate gene targeted by miR-196a (Figure 4A). The RT-qPCR and Western blotting results indicated that overexpression of miR-196a markedly diminished *CDKN1B* expression levels, whereas suppressing miR-196a elevated the relative abundance of *CDKN1B* expression (Figure 4B,C). We conducted dual-luciferase reporter assays to confirm the binding sites. These assays revealed that co-transfection with pmirGLO-WT and miR-196a mimic significantly reduced luciferase activity compared to co-transfection with pmirGLO-WT and miR-NC. Additionally, no significant difference in luciferase activity was observed between groups co-transfected with pmirGLO-MUT, irrespective of miR-196a mimic or miR-NC presence (Figure 4E). Furthermore, the Western blotting result indicated an increase in *CDKN1B* protein expression levels under heat stress, which was contrary to the expression trend observed for miR-196a under such conditions. These findings validate *CDKN1B* as a direct target gene of miR-196a. 

### 3.5. CDKN1B Reverses the Promotive Effect of miR-196a Overexpression on BMEC Proliferation

To investigate the role of *CDKN1B* as a target gene in miR-196a-mediated BMEC proliferation, BMECs were co-transfected with either a miR-196a mimic or miRNA-NC along with either a PCDNA or PCDNA-CDKN1B plasmid. The CCK-8 and EDU assays demonstrated that the OD value at 450 nm and the proportion of proliferating cells in the group co-transfected with the miR-196a mimic and PCDNA rose significantly compared to the miR-NC+PCDNA group (Figure 5A–C). This increase implies that miR-196a overexpression markedly enhances BMECs’ proliferative capacity. However, miR-196a’s promotion of BMEC proliferation was substantially negated when *CDKN1B* was overexpressed (Figure 5A–C). Furthermore, RT-qPCR data for proliferation-associated genes, namely, *PCNA*, *CDK2*, and *MCM3*, revealed an upregulation when the miR-196a mimic was overexpressed and a downregulation when cells were co-transfected with the miR-196a mimic and PCDNA-CDKN1B (Figure 5D–F). The Western blotting results indicated that overexpression of *CDKN1B* significantly increased its protein expression level. And overexpression of miR-196a significantly enhanced the protein expression level of *PCNA*, but this promoting effect was counteracted by *CDKN1B* (Figure 5G). These findings suggest that *CDKN1B* may reverse the proliferative effect of miR-196a on BMECs.

## 4. Discussion

HS is described as a physiological state where an animal cannot adequately dissipate heat, thereby disrupting its body’s thermal equilibrium [32]. Studies have documented that milk production diminishes when dairy cows are exposed to HS [33]. Reduced milk yield during HS can be attributed partly to nutritional intake and management considerations [34,35]. Furthermore, several studies have highlighted the significance of udder physiology status as a key determinant affecting milk production during HS [2,36]. In dairy cows, milk production hinges on the secretory activities of mammary epithelial cells [37]. HS adversely affects transcription, gene translation, and shifts in the BMEC cycle [1,38,39,40]. Thus, alleviating the inhibition of BMEC proliferation due to HS is pivotal for bolstering the productivity of dairy cows under such conditions.

A previous study emphasized significantly elevated miR-196a expression in the mammary gland compared to other tissues, which is consistent with our findings [26]. Therefore, miR-196a may play a crucial regulatory role in BMECs. To further investigate the potential role of miR-196a in BMECs’ response to HS, an HS model was established for BMECs. The upregulation of heat shock proteins (HSPs) serves as a well-established biomarker for detecting heat stress in animals [41]. The expression levels of *HSP90AA1* and *HSPB1* in BMECs under HS were evaluated. We also noticed a significant decrease in miR-196a expression in BMECs under HS exposure. Previously, miR-196a was proven to enhance cancer cell proliferation by targeting *FOXO1* and *HOXA5* [6,7,8]. Thus, miR-196a may play a role in cell proliferation under HS. Interestingly, miR-196a overexpression appeared to alleviate HS-induced proliferation inhibition.

mRNA-seq was used to gain preliminary insights into the potential functional roles of miR-196a in BMECs. The mRNA-seq analysis yielded 105 differentially expressed genes based on the thresholds of padj < 0.05 and |log2foldchange| ≥ 1. *MMP3*, *EDN1*, and *FGF21* exhibited significant upregulation, while *FOXO1*, *GADD45A*, and *SOCS2* showed significant downregulation. These genes are known to be closely linked to cellular proliferation [42,43,44,45,46,47]. The identified DEGs were categorized using the GO and KEGG databases. In the GO enrichment analysis, most of the DEGs were ascribed to GO terms associated with categories such as immune response, ion transport, mitochondrial inner membrane, and receptor regulator activity.

Within the top 20 KEGG pathways, seven pathways were associated with regulating cell proliferation. These pathways included the calcium signaling pathway, cAMP signaling pathway, p53 signaling pathway, JAK-STAT signaling pathway, FoxO signaling pathway, TNF signaling pathway, and transcriptional misregulation in cancer. All were found to be significantly enriched. Numerous studies have identified that the activation or suppression of these signaling pathways can either enhance or inhibit cell proliferation [27,28,29,30,48,49]. The enrichment analysis results suggest that miR-196a may play a role in regulating the proliferation of BMECs. However, the specific regulatory mechanisms through which miR-196a modulates cell proliferation via these pathways remain unknown and require further elucidation through future experiments. 

Several DEGs, such as *ATP5ME*, *NDUFAF8*, and *FGF21*, were significantly enriched in the thermogenesis pathway. These genes are established participants in controlling mitochondrial energy metabolism [44,50,51]. Given that many studies have posited their connection to cell thermogenesis [52], miR-196a may play a role in thermal balance regulation in BMECs during HS. The intricate underlying mechanisms require more detailed exploration.

To investigate miR-196a’s role in BMEC proliferation, we assessed cell viability and proliferation capabilities using CCK-8 and EDU assays. Both assay outcomes revealed that cells transfected with miR-196a mimic exhibited a significant increase in proliferation compared to cells transfected with miR-NC. Conversely, suppressing miR-196a dampened BMECs’ proliferation. Corroborating this, Shaoxuan et al. [10] found that miR-196a overexpression markedly improved the proliferation of immature porcine Sertoli cells. miR-196a inhibition notably curtailed their proliferation capacity, consistent with our findings. A separate study on laryngeal cancer further underscored miR-196a’s role in modulating cell proliferation [53].

One common method to evaluate cell proliferation is to monitor the expression of proliferation-associated genes. Proteins such as the proliferating cell nuclear antigen (*PCNA*) and minichromosome maintenance (*MCM*) are conventional markers for cell proliferation, expressed solely during cell division [54]. Another notable protein, cyclin-dependent kinase 2 (*CDK2*), belongs to the cell cycle protein kinase family and promotes cell proliferation when activated [55]. The RT-qPCR data indicated a significant upregulation of these genes in the group treated with miR-196a mimic. Suppressing miR-196a led to a pronounced downregulation of these genes.

To understand the mechanism through which miR-196a enhances BMEC proliferation more accurately, we integrated a list of downregulated genes based on the thresholds of padj < 0.05 and log2foldchange < 0 identified by mRNA-seq with findings from TargetScan Human 8.0 and miRDB. *CDKN1B* emerged as a potential target gene of miR-196a in this integrated analysis, especially given the significant role of *CDKN1B* in cell proliferation [15]. Subsequent tests using RT-qPCR, Western blotting, and dual-luciferase reporter assays confirmed this relationship, establishing *CDKN1B* as a target gene of miR-196a. To confirm the role of *CDKN1B* as a target gene influencing the effect of miR-196a on BMEC proliferation, BMECs were co-transfected with mimic, miR-NC, PCDNA, and PCDNA-CDKN1B vector. We determined that the proliferation-enhancing effect of miR-196a on BMECs could be reversed by *CDKN1B* through RT-qPCR, Western blotting, and EDU assay. These results suggest that miR-196a promotes BMEC proliferation by targeting *CDKN1B*. *CDKN1B* is a member of the Cip/Kip family and plays a crucial role as a cell cycle regulator, primarily involved in inhibiting or decelerating cell division [56,57]. *CDKN1B* can induce cell cycle arrest specifically in the G0/G1 phase [58]. In the normal progression of the cell cycle, *CDKN1B* levels are elevated during the G0/G1 phase. However, upon mitogenic stimulation, *CDKN1B* undergoes rapid degradation, facilitating the activation of Cdk2/cyclinE and Cdk2/cyclinA complexes to promote further progression through the cell cycle [59]. miR-196a can facilitate CDK activity and promote the proliferation of BMECs by downregulating *CDKN1B*. Past studies have revealed that one miRNA can influence the expression of various genes and, conversely, a single gene can be modulated by multiple miRNAs. Previous evidence also suggests that miR-196a promotes cell proliferation by targeting *FOXO1* and *HOXA5* [6,8]. The interaction between miR-196a and *CDKN1B* may play a crucial role in regulating BMEC proliferation.

Our study revealed that HS can reduce miR-196a expression and that increasing miR-196a levels can alleviate HS-induced suppression of cell proliferation. In our research, we identified *CDKN1B* as a target of miR-196a. Interestingly, several groups have reported that *CDKN1B* is directly involved in the cellular response to HS [16,17]. Given the established significance of the miR-196a and *CDKN1B* interaction in regulating BMEC proliferation, it would be worth investigating whether miR-196a mediates cell proliferation responses to HS via this specific pathway. These findings not only provide a foundation for further comprehensive study of miR-196a function in BMECs but also offer a novel insight into identifying potential targets for alleviating heat-stress-induced proliferation arrest in BMECs (Figure 6).

## 5. Conclusions

Our study findings indicate that HS reduces miR-196a expression and that overexpressing miR-196a can alleviate the HS-induced inhibitory effect of BMEC proliferation. Additional research confirms that miR-196a can promote BMEC proliferation by suppressing *CDKN1B* expression. These results could provide an important theoretical basis for understanding the function of miR-196a in BMECs and offer a novel insight into identifying potential targets for alleviating HS-induced proliferation arrest in BMECs.

## Figures and Tables

**Figure 1 animals-13-03682-f001:**
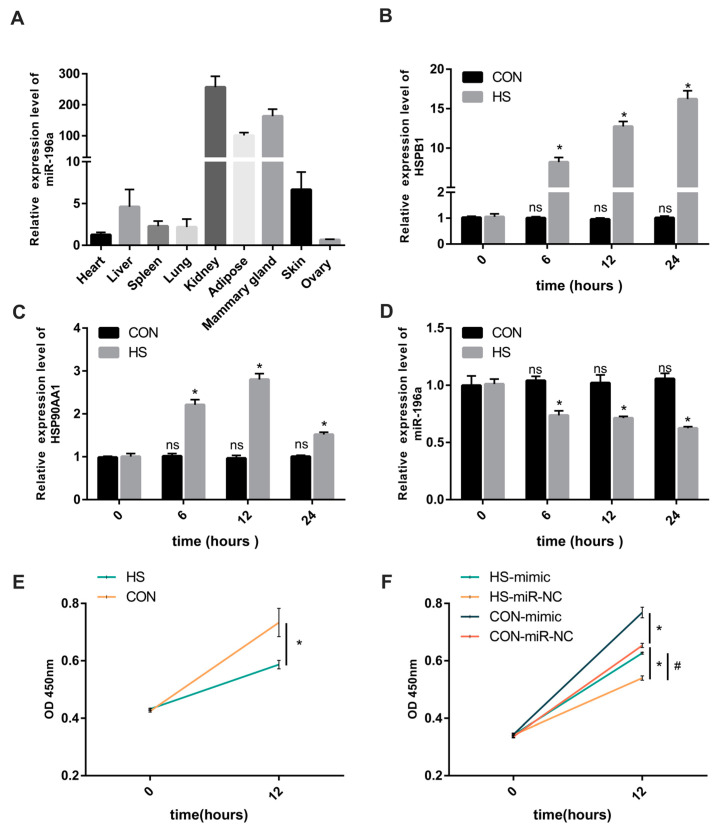
HS downregulates miR-196a expression and suppresses BMEC proliferation. (**A**) miR-196a expression in cow tissues was analyzed by RT-qPCR (*n* = 9). (**B**,**C**) The relative expression levels of *HSPB1* and *HSP90AA1* were measured after exposure to 42 °C for 0, 6, 12, and 24 h (*n* = 9). (**D**) The variation in miR-196a expression level was determined by RT-qPCR under HS (*n* = 9). (**E**) Cell viability of BMECs exposed to HS was measured by the CCK-8 assay (*n* = 5). (**F**) The effect of miR-196a on BMEC viability was measured by the CCK-8 assay (*n* = 5). * *p* < 0.05, vs. CON-miR-NC group, # *p* < 0.05, vs. HS-miR-NC group. The data are presented as means ± SEM. * *p* < 0.05, ns (no significance).

**Figure 2 animals-13-03682-f002:**
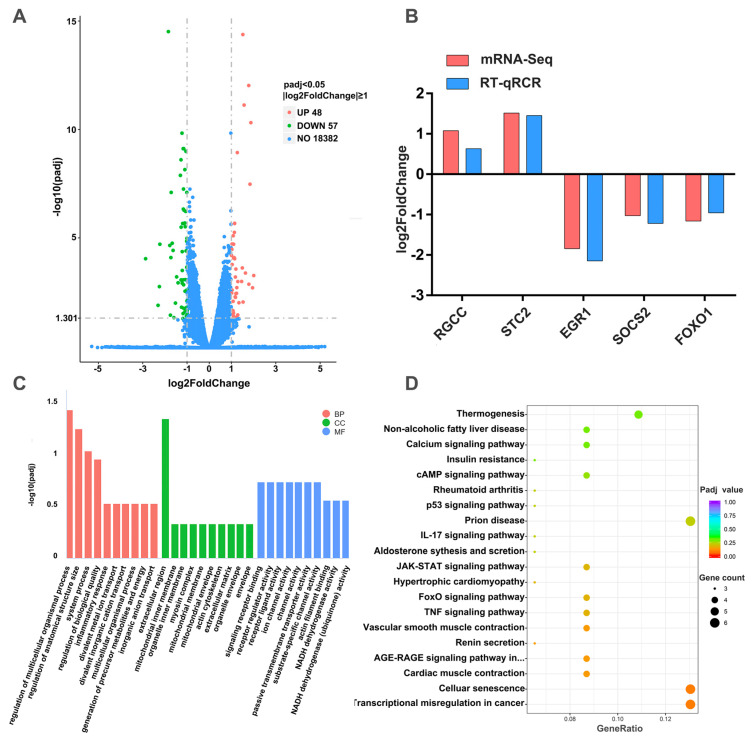
miR-196a’s potential functions were explored by mRNA-seq and bioinformatics analysis. (**A**) The volcano plot displays DEGs based on padj < 0.05, |log2foldchange| ≥ 1. (**B**) Validation of mRNA-seq results using RT-qPCR (*n* = 9). (**C**) Top 10 GO terms from enrichment analysis results of DEGs with padj < 0.05. (**D**) Top 20 significantly enriched KEGG pathways of DEGs with padj < 0.05.

**Figure 3 animals-13-03682-f003:**
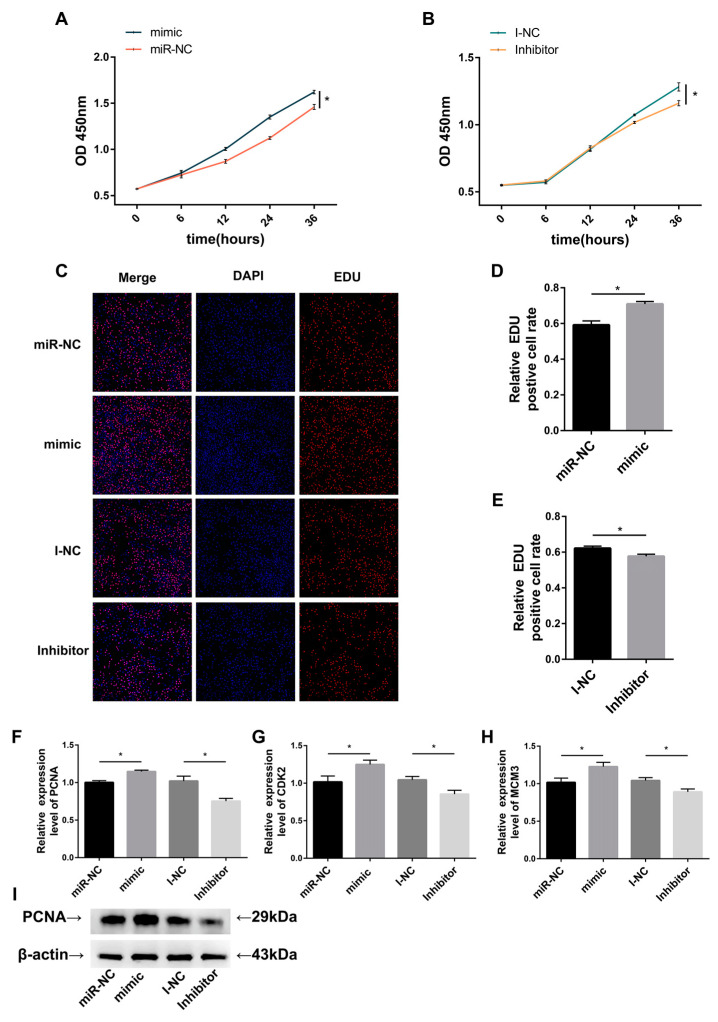
Effect of miR-196a on BMEC cell proliferation. (**A**,**B**) We conducted a CCK-8 assay to detect the viability of BMECs transfected with a miR-196a mimic or miR-196a inhibitor, respectively (*n* = 5). (**C**–**E**) An EDU assay was performed to measure the ratio of proliferating cells after overexpression or suppression of miR-196a, respectively (*n* = 5). (**F**–**H**) Gene transcription levels for *PCNA*, *CDK2*, and *MCM3* in BMECs were analyzed by RT-qPCR, with *β-actin* used as a reference (*n* = 9). (**I**) *PCNA* protein expression was analyzed by Western blotting (*n* = 3). *β-actin* was used as a reference. The data are presented as means ± SEM. * *p* < 0.05.

**Figure 4 animals-13-03682-f004:**
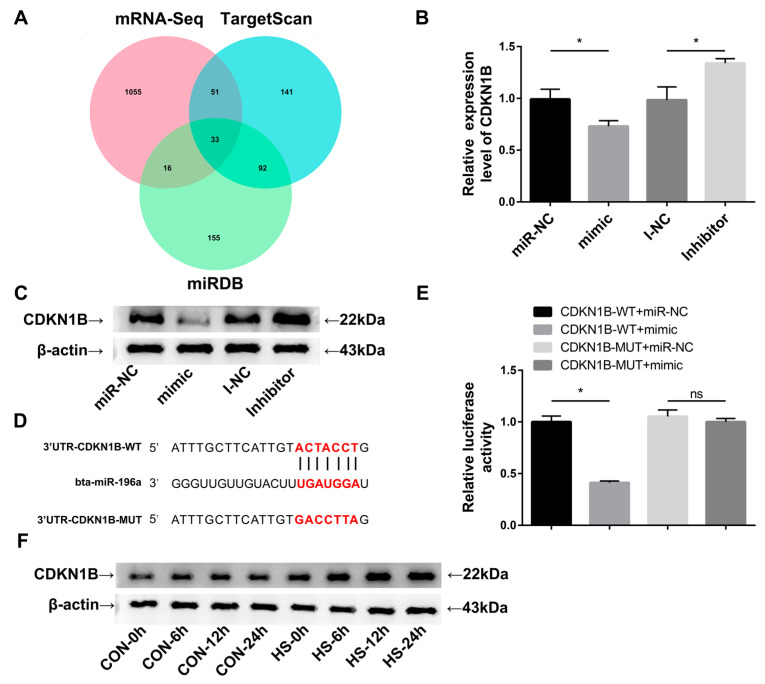
*CDKN1B* is a direct target gene of miR-196a in BMECs. (**A**) Venn diagram of target genes predicted by TargetScan Human 8.0 and miRDB overlapping with mRNA-seq downregulated genes (padj < 0.05, log2foldchange < 0). (**B**) The relative abundance of mRNA in the *CDKN1B* gene was measured by RT-qPCR (*n* = 9). (**C**) *CDKN1B* protein expression was examined by Western blotting. *β-actin* was used as an internal reference (*n* = 3). (**D**) The wild-type (WT) or mutant-type (MUT) 3′UTR fragments of *CDKN1B* were inserted into dual-luciferase reporter vectors, respectively. (**E**) The luciferase reporter assay was performed to verify the association between miR-196a and *CDKN1B* (*n* = 9). (**F**) Western blotting was performed to examine the expression level changes of *CDKN1B* under control and heat stress conditions. *β-actin* was used as an internal reference. The data are presented as means ± SEM. * *p* < 0.05, ns (no significance).

**Figure 5 animals-13-03682-f005:**
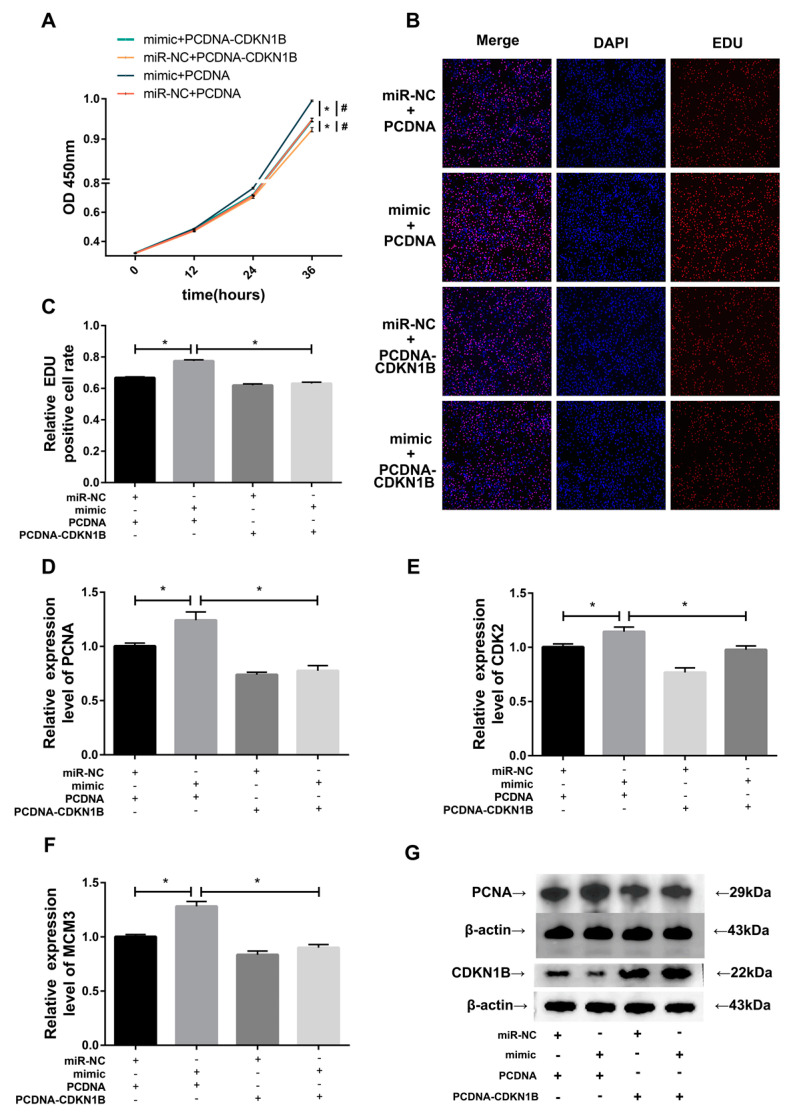
*CDKN1B* reverses the enhancing effect of miR-196a overexpression on BMEC proliferation. (**A**) BMEC viability was assessed using the CCK-8 assay after co-transfection (*n* = 5). * *p* < 0.05, compared to the miR-NC+PCDNA group; # *p* < 0.05, compared to the mimic+PCDNA-CDKN1B group. (**B**,**C**) The EDU assay was performed to measure the ratio of proliferating cells in BMECs transfected with miR-NC+PCDNA, miR-196a mimic+PCDNA, miR-NC+PCDNA-CDKN1B, and miR-196a mimic+PCDNA-CDKN1B (*n* = 5). (**D**–**F**) The relative abundance of proliferation-related genes, including *PCNA*, *CDK2*, and *MCM3*, were analyzed by RT-qPCR (*n* = 9). (**G**) The *CDKN1B* and *PCNA* protein expression levels was determined by Western blotting (*n* = 3). *β-actin* was used as an internal reference. The data are presented as means ± SEM. * *p* < 0.05.

**Figure 6 animals-13-03682-f006:**
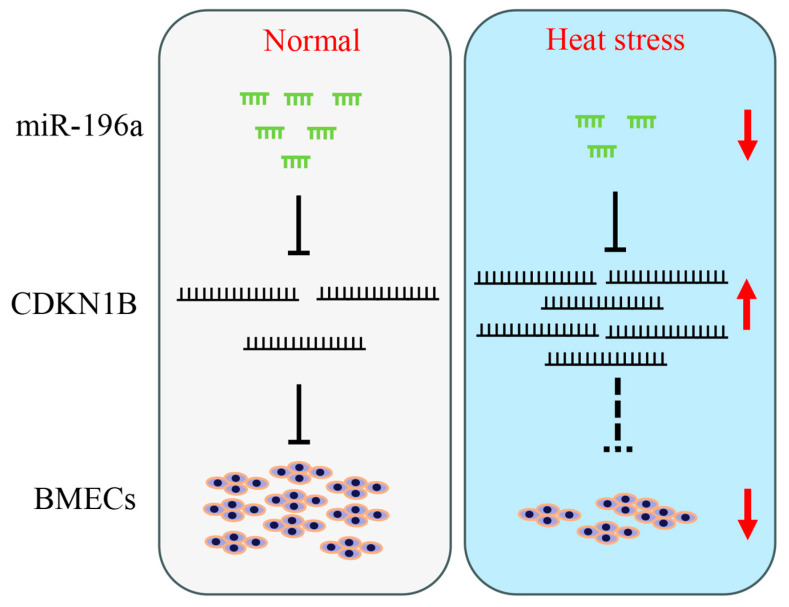
The potential mechanism of miR-196a alleviates cell proliferation inhibition caused by HS. miR-196a can promote BMEC proliferation by targeting *CDKN1B* at normal temperatures. miR-196a expression is downregulated during HS, which inhibits BMEC proliferation. However, increasing miR-196a levels can alleviate the suppressive effects of HS on cell proliferation. This process may occur through inhibition of *CDKN1B* expression by miR-196a. Solid lines indicate established regulatory mechanisms; dashed lines represent potential regulatory mechanisms. The upper arrow represents facilitation and the lower arrow represents inhibition.

## Data Availability

The mRNA-seq datasets generated during the current study are available in NCBI SRA (PRJNA995055). Data are contained within the article.

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
