# Peer review of "miR-196a Promotes Proliferation of Mammary Epithelial Cells by Targeting CDKN1B"

_animals, 2023, doi:10.3390/ani13233682_

Round 1

Reviewer 1 Report

Comments and Suggestions for Authors

The manuscript entitled "miR-196a Promotes Proliferation of Mammary Epithelial Cells by Targeting CDKN1B" by Chen et al., the experimental design and scientific soundness are well, and the writting and presention are also well, the manuscript can be accept after minor revision.

1. Title: adding "heat stress" in the title.

2. References: the cited references in the text and lists should according to the format style requirement of the journal.

3. Introduction: simplify the introduction section, focus on the research progress in animals.

Comments on the Quality of English Language

Minor editing of English language required

Reviewer 2 Report

Comments and Suggestions for Authors

just minor changes:

Line 35: sort key words by alphabetic order. Key words cannot be the same works you are using as title.

Line 79-85: it is a mixture of objectives and results. Please, clarify your objectives.

Line 250: write Bos taurus as Bos taurus

Line 264: change “we conducted GOand KEGG” to “we conducted GO and KEGG”

Line 266: change “Figure 2C showcases” to “Figure 2C show cases”

Reviewer 3 Report

Comments and Suggestions for Authors

The aim of this paper is to investigate the role of miR-196a in protecting Bovine Mammary Epithelial Cells (BMECs) against heat stress (HS)-induced damage and to understand the underlying mechanism. The main contributions of the study are the discovery that HS inhibits BMEC proliferation and down-regulates miR-196a expression, while overexpression of miR-196a counteracts this inhibition. The paper also identifies CDKN1B as a target gene of miR-196a and suggests that the miR-196a/CDKN1B axis may play a crucial role in alleviating heat stress-induced proliferation arrest in BMECs. The strengths of the paper include the experimental evidence supporting these findings and the potential implications for improving udder health in dairy cows under heat-stress conditions. The analysis is well-executed, utilizing a robust dataset, and the study design is commendable, providing valuable insights. However, there are certain areas that should be addressed to enhance the manuscript's potential for publication.

Comments:

Point 1: Kindly ensure that you provide proper references when mentioning the software or R packages used in the bioinformatics analysis section (Section 2.5).

Point 2: On page 5, in line 207, there is no need to include an asterisk (*) before the p-value.

Point 3: The conclusions section in this manuscript requires improvement to provide a clearer explanation of the study's results, its significance, and the implications for future research.

Reviewer 4 Report

Comments and Suggestions for Authors

All comments are enclosed in the attached PDF 

kindest regards

Comments on the Quality of English Language

See pdf enclosed (I will send it to the editor since the system does not allow me to enclose a PDF).

Round 2

Reviewer 3 Report

Comments and Suggestions for Authors

Dear Authors, I have reviewed your manuscript and commend the revisions made according to the comments and suggestions provided during the first revision. I appreciate your thorough attention to detail in addressing the comments, which significantly improved the quality and clarity of the manuscript. Based on the revisions, I am pleased to accept the article for publication.

Reviewer 4 Report

Comments and Suggestions for Authors

Dear authors and editor,

I have reread the article and think it is now acceptable for publication.

I only invite the authors to modify the supplemetary material with the full images of the western blots (they still reflect the older versions).

All the best